# Nardochinoid B Inhibited the Activation of RAW264.7 Macrophages Stimulated by Lipopolysaccharide through Activating the Nrf2/HO-1 Pathway

**DOI:** 10.3390/molecules24132482

**Published:** 2019-07-06

**Authors:** Yun-Da Yao, Xiu-Yu Shen, Jorge Machado, Jin-Fang Luo, Yi Dai, Chon-Kit Lio, Yang Yu, Ying Xie, Pei Luo, Jian-Xin Liu, Xin-Sheng Yao, Zhong-Qiu Liu, Hua Zhou

**Affiliations:** 1Faculty of Chinese Medicine, Macau University of Science and Technology, Taipa, Macao 999078, China; 2State Key Laboratory of Quality Research in Chinese Medicine, Macau University of Science and Technology, Taipa, Macao 999078, China; 3College of Traditional Chinese Materia Medica, Shenyang Pharmaceutical University, Shenyang 110016, China; 4ICBAS-Laboratory of Applied Physiology, Abel Salazar Institute of Biomedical Sciences, University of Porto, Rua de Jorge Viterbo Ferreira, 228, 4050-313 Porto, Portugal; 5Institute of Traditional Chinese Medicine and Natural Products, College of Pharmacy, Jinan University, Guangzhou 510632, China; 6College of Pharmacy, Hunan University of Chinese Medicine, Changsha 418000, China; 7Joint Laboratory for Translational Cancer Research of Chinese Medicine of the Ministry of Education of the People’s Republic of China, Guangzhou University of Chinese Medicine, Guangzhou 510006, China

**Keywords:** *Nardostachys chinensis*, nardochinoid B, nitric oxide, inducible nitric oxide synthase, heme oxygenase-1

## Abstract

Nardochinoid B (NAB) is a new compound isolated from *Nardostachys chinensis*. Although our previous study reported that the NAB suppressed the production of nitric oxide (NO) in lipopolysaccharide (LPS)-activated RAW264.7 cells, the specific mechanisms of anti-inflammatory action of NAB remains unknown. Thus, we examined the effects of NAB against LPS-induced inflammation. In this study, we found that NAB suppressed the LPS-induced inflammatory responses by restraining the expression of inducible nitric oxide synthase (iNOS) proteins and mRNA instead of cyclooxygenase-2 (COX-2) protein and mRNA in RAW264.7 cells, implying that NAB may have lower side effects compared with nonsteroidal anti-inflammatory drugs (NSAIDs). Besides, NAB upregulated the protein and mRNA expressions of heme oxygenase (HO)-1 when it exerted its anti-inflammatory effects. Also, NAB restrained the production of NO by increasing HO-1 expression in LPS-stimulated RAW264.7 cells. Thus, it is considered that the anti-inflammatory effect of NAB is associated with an induction of antioxidant protein HO-1, and thus NAB may be a potential HO-1 inducer for treating inflammatory diseases. Moreover, our study found that the inhibitory effect of NAB on NO is similar to that of the positive drug dexamethasone, suggesting that NAB has great potential for developing new drugs in treating inflammatory diseases.

## 1. Introduction

Inflammation is a kind of defensive reaction of living organisms with vascular systems to harmful factors such as pathogens, damaged cells, and irritants [1]. The inflammation may happen in a number of diseases, such as arthritis, arthrophlogosis, asthma, and so on [2]. The nonsteroidal anti-inflammatory drugs (NSAIDs) are widely used for fighting against inflammation diseases in clinical conditions. Restraining the cyclooxygenases (COXs) to inhibit prostaglandins is the main mechanism by which NSAIDs produce their anti-inflammatory effect [3]. However, many critical side effects, such as the increasing risk of serious and even fatal stomach and intestinal adverse reactions [4], myocardial infarction [5], stroke [6], systemic and pulmonary hypertension [7], and heart failures [8], happen during COX inhibition. Therefore, NSAIDs are not ideal for treating every inflammatory disease because of their side effects in the clinic. Thus, it is necessary to develop new, safer drugs to treat inflammation diseases better.

Macrophages play important roles in the innate immune response. They protect cells from injury induced by exogenous factors such as bacteria and viruses and endogenous factors such as other damaged cells. Also, macrophages promote the repair processes of tissue injury [9]. Proinflammatory mediators, such as interleukin-1β (IL-1β), interleukin-6 (IL-6), tumor necrosis factor alpha (TNF-α), prostaglandin E_2_ (PGE_2_), and nitric oxide (NO) [10,11,12,13,14], are produced by the activated macrophages and then promote the development of inflammation [15]. Thus, in our study, the LPS-stimulated RAW264.7 cells, a classical inflammatory cell model [16], was chosen to study the anti-inflammatory mechanism of NAB.

In recent years, there has been growing interest in the anti-inflammatory effects of natural components present in commonly used traditional herbal medicines. *Nardostachys chinensis* is one of the traditional Chinese medicines that was reported to have an anti-inflammatory effect [17]. The extracts of the plant roots and rhizomes of *N. chinensis* have been used for the treatment of blood disorders, disorders of the circulatory system, and herpes infection [18]. Recently, some compounds isolated from *N. chinensis* were reported to inhibit the protein expression of inducible nitric oxide synthase (iNOS) and cyclooxygenase-2 (COX-2) in LPS-activated RAW264.7 macrophages [18,19,20]. Nardochinoid B (NAB) is a compound isolated from *N. chinensis*. Our previous research has proved that NAB inhibits the production of NO in the LPS-induced RAW264.7 macrophages [20]. However, the mechanisms of the anti-inflammatory action of NAB have not been identified clearly. In this study, the mechanisms of anti-inflammatory activity and the antioxidant effect of nardochinoid B (NAB) were for the first time investigated in LPS-stimulated RAW264.7 cells.

The progression of inflammation could be inhibited through activating the nuclear factor erythroid 2-related factor 2 (Nrf2) pathway, meaning that activating the Nrf2 pathway could be a potential therapeutic strategy in anti-inflammatory disorders [21]. The translocation of Nrf2 protein into the cell nucleus induces the expression of heme oxygenase (HO)-1. Then, followed by the overexpression of HO-1, the production of inflammatory mediators is reduced and the inflammatory process is modulated [22]. Yet, few Nrf2 activators have been validated and used in the clinic. Tecfidera (dimethyl fumarate) is one of the Nrf2 activators that have been approved for the treatment of multiple sclerosis [23]. However, the long-term use of it causes several side effects [24]. Therefore, the discovery of new, safer Nrf2 activators for the clinic has become an essential and urgent matter.

In the present study, we have focused on these certain aspects of NAB: (1) whether NAB has the ability to suppress the LPS-induced inflammatory responses in RAW264.7 cells, and (2) whether NAB upregulates HO-1 to promote its anti-inflammatory effects by activating the Nrf2 signaling pathway. The results in this study revealed that NAB exerted its anti-inflammatory effects in LPS-induced RAW264.7 cells in a manner related to the activation of the Nrf2/HO-1 pathway, rather than the inhibition of the nuclear factor-κB (NF-κB) pathway and mitogen-activated protein kinase (MAPK) pathway.

## 2. Results

### 2.1. Anti-Inflammatory Activities of NAB on LPS-Activated RAW264.7 Macrophages

#### 2.1.1. NAB Reduced the Release of NO in LPS-Stimulated RAW264.7 Macrophages

The results from the MTT assay show that NAB (Figure 1) had no significant cytotoxicity to LPS-stimulated RAW264.7 cells at the concentrations lower than 20 μM (Figure 2A,B). The nitrite level (evaluated through the stable oxidized product of NO) and the production of PGE_2_ in the culture medium of the RAW264.7 cells were significantly increased (*P* < 0.01) after 18 h of LPS stimulation. The pretreatment with NAB markedly decreased the LPS-induced NO production in a concentration-dependent manner (Figure 2C), while it did not inhibit the production of PGE_2_ (Figure 2D). Dexamethasone (DEX) was selected to serve as the positive control. The results show that DEX markedly reduced the production of both NO and PGE_2_ in LPS-stimulated RAW264.7 cells (Figure 2C,D).

#### 2.1.2. NAB Inhibited the Expression of iNOS Rather Than COX-2 in LPS-Stimulated RAW264.7 Macrophages

The mRNA and protein expression of iNOS and COX-2 in the cells were significantly increased after stimulation with LPS (100 ng/mL) for 18 h (Figure 3). NAB markedly downregulated the protein expression level of iNOS in the LPS-stimulated RAW264.7 cells in a concentration-dependent manner (Figure 3A) and decreased the mRNA expression of iNOS at the concentration of 10 μM (Figure 3C). However, NAB did not significantly downregulate the mRNA and protein expression levels of COX-2 in the same conditions (Figure 3B,D).

### 2.2. Potential Anti-Inflammatory Mechanisms of NAB on LPS-Induced RAW264.7 Macrophages

#### 2.2.1. NAB Increased the mRNA and Protein Expression Levels of HO-1 in LPS-Stimulated RAW264.7 Cells

The results (Figure 4) show that the expression level of HO-1 was increased after stimulation with LPS (100 ng/mL) for 6 h. Sulforaphane (SFN), a confirmed Nrf2 activator, was selected as an alternate positive control drug to DEX in the following mechanism study. NAB significantly increased the mRNA (Figure 4A) and protein (Figure 4B) expression levels of HO-1 in LPS-stimulated RAW264.7 cells.

#### 2.2.2. NAB Promoted Nrf2 Protein Translocation into the Nucleus in RAW264.7 Macrophages

As shown in Figure 5, the pretreatment of NAB promoted Nrf2 protein entering the nucleus in RAW264.7 cells, similar to the effect of SFN.

#### 2.2.3. NAB Suppressed the Production of TNF-α, IL-1β, and IL-6

The results show that the LPS stimulation of RAW264.7 cells increased the expression levels of TNF-α (Figure 6A), IL-1β (Figure 6C), and IL-6 (Figure 6E) in the culture medium. The mRNA expression levels of TNF-α (Figure 6B), IL-1β (Figure 6D), and IL-6 (Figure 6F) were induced by LPS as well. The treatment of NAB downregulated the expression levels of TNF-α (Figure 6A,B), IL-1β (Figure 6C,D), and IL-6 (Figure 6E,F). The positive control drug sulforaphane also significantly inhibited the expression level of these inflammatory mediators (Figure 6).

#### 2.2.4. NAB Failed to Inhibit the Activation of the NF-κB and MAPK Pathways in LPS-Stimulated RAW264.7 Cells

As shown in Figure 7, the LPS stimulation of RAW264.7 cells increased the expression levels of phospho-p65 (p-p65), phospho-p38 (p-p38), and phospho-extracellular regulated protein kinase (p-ERK). However, NAB failed to inhibit the increased expression levels of p-p65 (Figure 7A), p-ERK (Figure 7B), and p-p38 (Figure 7C).

## 3. Discussion

As described before, macrophages play an important role in inflammation as they are able to release different kinds of cytokines to ignite inflammatory reactions [9]. The LPS-stimulated RAW264.7 macrophages is a kind of classical inflammatory cell model widely used in evaluating the anti-inflammatory effect and mechanisms of many natural products derived from Chinese medicines [25]. Therefore, we chose the LPS-stimulated RAW264.7 cells as the cell model in this study. Dexamethasone (DEX) and sulforaphane (SFN) were chosen as the positive control drugs in this study. DEX is a classic anti-inflammatory drug that is widely used in the clinic [26]. It is a steroidal anti-inflammatory drug that has been widely used to treat rheumatoid arthritic knees [9], pneumonia [27], and bronchiolitis [11]. SFN is a kind of drug that has been confirmed as a Nrf2 activator. It is a natural isothiocyanate, and it has been proved that SFN could suppress LPS-induced inflammation in mouse peritoneal macrophages through activating the Nrf2 pathway and upregulating the HO-1 expression [28]. Moreover, SFN inhibited the expression of some inflammatory mediators, including TNF-α, IL-1β, and IL-6 [29], through activating the Nrf2 pathway. So, DEX was chosen as the positive control drug in the study to evaluate the anti-inflammatory activity of NAB, and SFN was chosen as the positive control in the study to evaluate the Nrf2 pathway-related mechanism of NAB.

In this study, we firstly evaluated the cytotoxicity of NAB and found that NAB had no significant cytotoxicity to LPS-stimulated RAW264.7 cells at the concentrations lower than 20 μM (Figure 2A,B). Thus, we selected the concentrations of NAB ranging from 2.5 μM to 10 μM to examine the anti-inflammatory activity of NAB in the LPS-stimulated RAW264.7 cells. Then, following evaluation of the effect of NAB on the production of NO and PGE_2_ by the LPS-induced RAW264.7 cells, we examined the effect of NAB on the expression of iNOS and COX-2 by LPS-stimulated RAW264.7 macrophages, since iNOS and COX-2 are the enzymes responsible for the production of NO and PGE_2_, respectively. After that, the expression level of HO-1 in LPS-stimulated RAW264.7 cells was detected with the treatment of NAB, because HO-1 is one of the regulating factors of the expression of iNOS. As the translocation of Nrf2 protein into the cell nucleus mediates the expression of HO-1, the migration level of Nrf2 protein in the RAW264.7 macrophages was evaluated. Moreover, as the macrophages release cytokines (e.g., TNF-α, IL-1β, and IL-6) [30] to promote and encourage the development and progression of inflammation in vivo, these inflammatory mediators were detected in this study.

NO and PGE_2_ are two of the most important inflammatory mediators that participate in inflammatory processes. The inflammation and the exposure of tissue cells to bacterial products such as LPS, lipoteichoic acid (LTA), peptidoglycans, and bacterial DNA or whole bacteria will induce the high expression of iNOS and then enhance the production of NO. In these situations, the NO forms peroxynitrite, which acts as a cytotoxic molecule, resists invading microorganisms, and acts as a killer [31]. However, it has been reported that in aseptic inflammation, the iNOS expression and NO formation would also be induced in human macrophages; for example, in rheumatoid arthritis and osteoarthritis [32]. In these bacteria-free inflammatory processes, the synthesis of NO can be an important factor that helps maintain the inflammatory and osteolytic processes [13]. PGE_2_ mediates the increasing of arterial dilation and microvascular permeability. This action will cause blood to flow into the inflamed tissue and thus causes redness and edema [33]. COX-2 belongs to the regulatory enzymes involved in the production of PGE_2_, and it also regulates the synthesis of prostaglandin I_2_ (PGI_2_, also called as prostacyclin) and thromboxaneA_2_ (TXA_2_) [34]. It is known that TXA_2_ is the major cyclooxygenase product in platelets. It is also a potent vasoconstrictor and can stimulate the aggregation of platelets in vitro. PGI_2_ is produced and synthesized in vascular endothelial cells. It is a vasodilator and inhibitor of platelets [34]. It has been proven that the inhibition of COX-2 may break the balance between PGI_2_ and TXA_2_, leading to cardiovascular risks [35]. Fortunately, in this study, the results show that NAB only targets and inhibits NO and iNOS and does not affect the expression of COX-2 (Figure 2C,D and Figure 3). Thus, NAB may have low cardiovascular side effects compared with DEX. However, the results showed that the protein expression level of iNOS was inhibited by the concentration of 2.5 μM NAB, while the mRNA expression level was not; thus, it was considered that NAB may affect the translation process of iNOS from gene to protein.

TNF-α, IL-1β, and IL-6 belong to the inflammatory cytokines and are can also be involved in inflammatory processes [36]. In this research, all these inflammation cytokines and regulatory enzymes (iNOS and COX-2) were upregulated by the LPS stimulation (Figure 3 and Figure 6). Then, the increases of these inflammatory mediators were significantly inhibited by NAB (Figure 6). More importantly, the inhibitory effect of NAB on the mRNA expression level of TNF-α is better than that of SFN, meaning that NAB has obvious anti-inflammatory activity in LPS-stimulated RAW264.7 macrophage cells.

Usually, the inflammatory processes are accompanied by the activation of the NF-κB pathway, which also promotes the expression of inflammatory mediators in macrophages [37]. Previous research has shown that the production of inflammatory cytokines is related to the LPS-induced activation of the NF-κB pathway [38]. The mitogen-activated protein kinases (MAPK) pathway also plays an critical role in inflammatory responses [39]. The activation of both NF-κB and MAPK signaling pathways is involved in the development of inflammation [25]. Therefore, under normal circumstances, inhibiting NF-κB and MAPK signaling pathways is considered as an effective way to combat inflammatory reaction. The activation of NF-κB resulted in the phosphorylation of nuclear factor of kappa light polypeptide gene enhancer in B-cells inhibitor, alpha (IκBα), IκB kinase-α (IKKα), and p65, leading to the transcription of inflammatory genes and the expression of inflammatory proteins [40]. The activation of the MAPK pathway results in the phosphorylation of p38, c-Jun N-terminal kinase (JNK), and ERK [41], which may promote proinflammatory cytokine production [42]. Some reports showed that the deactivation of NF-κB and MAPK pathways in RAW264.7 cells leads to the inhibition of LPS-induced NO, PGE_2_, iNOS, COX-2, TNF-α, and IL-6 production [25,43]. Other studies have reported that the extracts of *N. chinensis* inhibited the p38 MAPK pathway to inhibit the expression of inflammatory mediators [44]. To study the anti-inflammatory mechanism of NAB, we first investigated the effect of NAB on the activation of the NF-κB and MAPK pathways in LPS-stimulated RAW264.7 cells. However, the results showed that NAB did not inhibit the activation of the NF-κB and MAPK pathways (Figure 7), so NAB may not act on the NF-κB and MAPK pathways to exert its anti-inflammatory effects.

The activation of the Nrf2 pathway is another possible way to prevent LPS-induced transcriptional upregulation of proinflammatory cytokines, including TNF-α, IL-1β, and IL-6 [45]. These inflammatory cytokines were decreased by the Nrf2-dependent antioxidant genes HO-1 and NQO-1. In Nrf2-knockout mice, the mRNA and protein levels of COX-2, iNOS, IL-6, and TNF-α increased [46] and the anti-inflammatory effect also disappeared [47]. Since the current result showed that NAB inhibited TNF-α, IL-1β, and IL-6 obviously (Figure 6), it was hypothesized that NAB may activate the Nrf2 pathway to exert its anti-inflammatory effect.

In this study, it was found that NAB inhibited the expression of the inflammatory protein iNOS (Figure 3A,C) and inflammatory cytokines including NO, TNF-α, and IL-6 (Figure 2C and Figure 6), accompanied by the increase of antioxidant protein HO-1 (Figure 4). More importantly, the study found that NAB had no inhibitory effect on COX-2 (Figure 3B,D) and PGE_2_ (Figure 2D), suggesting that NAB has potential to be developed as a selective iNOS/NO inhibitor, a kind of anti-inflammatory drug that helps to reduce airway inflammatory responses, such as the compound 1400W [48], and relieve the pain caused by mechanical damage, such as the compound AR-C102222 [49]. Further, since NAB did not affect the expressions of COX-2 and PGE_2_, it is safer than NSAIDs, which inhibit PGE_2_ to exert their anti-inflammatory effect through inhibiting COX-2 expression. At the same time, our study found that the inhibitory effect of NAB on NO is very similar to that of the positive control drug DEX, suggesting that NAB has great development value in future study.

Oxidative or nitrosative stress, cytokines, and other mediators may cause the cells to overproduce HO-1 to protect themselves [22,50]. The induction of HO-1 reduces the production of inflammatory mediators and modulates the inflammatory process [51]. HO-1 can be rapidly induced by various oxidative response-inducing agents, including LPS [22]. The current results also show that LPS increased the level of HO-1 slightly, but compared with the LPS group, NAB further increased the level of HO-1 protein dramatically (Figure 4). The NO production induced by LPS was inhibited by the high expression of HO-1 [52]. In this study, NAB reduced NO production while increasing HO-1 expression (Figure 4); the current results are consistent with the finding that the high expression of HO-1 can inhibit LPS-induced NO production [52].

Another factor that is related to the expression of HO-1 is the expression of interleukin (IL)-10. IL-10 induces the phosphorylation of Janus Kinase (Jak) 1 and the activation of signal transducer and activator of transcription (STAT)-1 and STAT-3 [53]. Also, IL-10 activates phosphatidylinositol-3 kinase (PI3K), which is involved in the proliferative effects of IL-10 [54]. It has been proven that IL-10 can induce the expression of HO-1 [55]. Also, it has been reported that the IL-10-induced activation of STAT-3 and PI3K was associated with the expression of HO-1 [56]. Thus, further study will focus on the role of NAB in the expression of IL-10 and the associated production such as STAT-1 and PI3K.

Taken together, the results suggest that the activation of the Nrf2/HO-1 pathway is the potential mechanism by which NAB exerts its anti-inflammatory effects against LPS-activated inflammation (Figure 8).

## 4. Materials and Methods

### 4.1. Materials

Nardochinoid B (NAB) (Figure 1) (HPLC purity >98%) was provided by the Institute of Traditional Chinese Medicine and Natural Products, Jinan University (Guangzhou, China). Dimethyl sulfoxide (DMSO) (Sigma, Cat. No. D2625, St. Louis, MO, USA) was used to dissolve the NAB powder to give a stock solution of 30 mM concentration. Lipopolysaccharide (LPS), dexamethasone (DEX), and sulforaphane (SFN) were purchased from Sigma Chemical Co. (St. Louis, MO, USA). Antibodies to iNOS [57], COX-2 [58], heme oxygenase (HO)-1 [58], Nrf2 [18], p-p65 [18], p-p38 [18], and phospho-extracellular regulated protein kinases (p-ERK) [18] were obtained from Cell Signaling Technology (Boston, MA, USA). Antibodies to β-actin and laminin B1 [18] were from Santa Cruz Biotechnology (Santa Cruz, CA, USA). Antibody to α-tubulin [18] was from Sigma Chemical Co. (St. Louis, MO, USA). The secondary antibodies for Western blot were from Li-COR Biotechnology (Lincoln, NE, USA). ELISA kits for IL-1β, IL-6, and TNF-α was from eBioscience (eBioscience, Inc., San Diego, CA, USA). ELISA kit for PGE_2_ was from Cayman Chemical (Cayman Chemical, Ann Arbor, MI, United States). The nitric oxide (NO) production level was measured by a Griess Reagent System kit, which was obtained from Promega Corporation (Madison, WI, USA).

### 4.2. Cell Culture

The immortalized mouse macrophage cell line RAW264.7 was obtained from the American Type Culture Collection (ATCC, Manassas, VA, USA). Dulbecco’s modified Eagle’s medium (DMEM) supplement with 10% heat-inactivated fetal bovine serum (FBS) (Gibco BRL Co, Grand Island, NY, USA), penicillin G (100 units/mL), streptomycin (100 mg/mL), and l-glutamine (2 mM) (Gibco BRL Co, Grand Island, NY, USA) was chosen to maintain the cells. The cells were incubated at 37 °C in a humidified atmosphere containing 5% CO_2_ and 95% air.

### 4.3. Cell Viability Assay

The cells were seeded in 96-well plates at the density of 1.4 × 10^4^ cells/well and were incubated for 24 h. After incubation, the cells were pretreated with different concentrations (1.25, 2.5, 5, 10, 20, and 40 μM) of NAB for 1 h. Then, the cells were stimulated with or without LPS (100 ng/mL) for 18 h. Cytotoxicity was analyzed by using MTT assay. MTT solution (5 g/L) was added to each well and incubated for 4 h at 37 °C. Then, 100 μL 10% sodium dodecyl sulfate (SDS)–HCl solution was added to the wells and incubated for another 18 h. The optical density was read at 570 nm (reference, 650 nm) using a microplate UV/VIS spectrophotometer (Tecan, Mannedorf, Switzerland). The control group, in which the cells were not treated with compounds and LPS, was set as 100% for its cell viability.

### 4.4. Determination of NO, PGE_2_, TNF-α, IL-1β, and IL-6 Production

The cells were plated in 24-well plates at the density of 8 × 10^4^ cells/well and were incubated for 24 h. Then, the cells were pretreated with different concentrations (2.5, 5, and 10 μM) of NAB and positive control drug (dexamethasone, DEX, or sulforaphane, SFN) for 1 h, respectively. LPS (100 ng/mL) was added to the culture medium and the cells were stimulated with LPS for another 18 h. After incubating the cells with drugs and LPS, the cells and the medium were collected and stored at −80 °C. NO production was measured as the nitrite concentration in the medium by the Griess reagent (Promega, Madison, WI, USA). The TNF-α, IL-1β, and IL-6 concentrations in the culture medium were measured by using the enzyme-linked immunosorbent assay (ELISA) kit (eBioscience, Inc., San Diego, CA, USA), and the PGE_2_ concentration in the cell supernatant was detected by the ELISA kit from Cayman Chemical (Cayman Chemical, Ann Arbor, MI, United States).

### 4.5. Protein Preparation and Western Blot Analysis

The cells in 24-well plates were collected after being treated with drugs and LPS for 6 h (for HO-1 proteins) or 18 h (for other inflammation-related proteins). RIPA lysis buffer (Cell Signaling technology, Boston, MA, USA) was mixed with 1× protease inhibitor (Roche Applied Science, Mannheim, Germany) and the mixture was used to lyse the collected cells to extract total protein. For the measurement of Nrf2 protein, cells were treated with NAB (10 μM) and SFN (5 μM) for 6 h, and then the NE-PER Nuclear and Cytoplasmic Extraction Reagents (Thermo Scientific, Rockford, IL, USA) were used to extract the cytoplasmic and nuclear extracts. The protein concentration was determined with the Bio-Rad Protein Assay (Bio-Rad, Hercules, CA, USA). Thirty micrograms of these protein samples was resolved by 6% (for Nrf2 measurement), 10% (for iNOS, COX-2, p-p65, p-ERK, and p-p38 measurements), and 12% (for HO-1 measurement) sodium dodecyl sulfate polyacrylamide gel electrophoresis (SDS-PAGE). After electrophoresis separation, the proteins were transferred from the gel onto nitrocellulose membrane (GE Healthcare Life Sciences, Buckinghamshire, UK). Then, the membrane was blocked with 5% skimmed milk and then incubated with the primary antibodies (iNOS, COX-2, HO-1, Nrf-2, p-ERK, p-p65, and p-p38) and mouse antibodies specific for β-actin (for iNOS, COX-2, HO-1, and p-p65 measurements), α-tubulin (for p-ERK and p-p38 measurements), and laminin B1 (for Nrf2 measurement) at 4 °C overnight. After that, the membrane was incubated with IRDye 800CW goat anti-mouse IgG (H + L) or IRDye 800CW goat anti-rabbit IgG (H + L) secondary antibodies (Li-COR, Lincoln, NE, USA) at room temperature for 1 h. The antigen–antibody complex bands were examined with an Odyssey CLxImager (Li-COR, Lincoln, NE, USA) and the protein expression level was quantified by using Odyssey v3.0 software (Li-COR, USA). The density ratios of iNOS, COX-2, HO-1, Nrf-2, p-ERK, p-p65, and p-p38 to β-actin, α-tubulin, or laminin B1 were calculated for evaluating the anti-inflammatory effect and underlying mechanism of NAB.

### 4.6. RNA Extraction and Quantitative Real-Time Polymerase Chain Reaction (qRT-PCR)

The cells in 24-well plates were collected after being treated by tested drugs and LPS for 6 h (for the HO-1 test) or 18 h (for iNOS, COX-2, TNF-α, IL-1β, and IL-6 tests). Total RNA was isolated from cells with the NucleoSpin RNA kit (Macherey-Nagel, Düren, Germany). The total RNA concentration for each sample was detected by using a NanoDrop spectrophotometer (Thermo Scientific, USA). One microgram of total RNA of each sample were used for reverse transcription into cDNA by using the reverse transcription Universal cDNA Master Kit (Roche Applied Science, Germany). Target RNA levels were determined by using ViiATM 7 real-time PCR, where 1 μL cDNA, 2 μL primers, 10 μL SYBER Green PCR Master Mix (Roche, Mannheim, Germany), and 7 μL PCR-grade water were used in the PCR reaction. The denaturation step of the PCR reactions was set to 95 °C for 10 min. Forty cycles were repeated at 95 °C for 15 s and 60 °C for 1 min. The 2^−∆∆Ct^ cycle threshold method was used to normalize the relative mRNA expression levels to the internal control. The primers used in this study are listed in Table 1.

### 4.7. Data Analysis

All data are presented as the mean ± SEM of three independent experiments. The statistical analyses for these results were carried out with GraphPad Prism 7 (GraphPad Software, San Diego, CA, USA) by using one-way ANOVA followed by post-hoc analysis with Tukey’s multiple comparison test to compare the difference between groups. In all cases, a level of *P* < 0.05 was considered statistically significant.

## 5. Conclusions

From the above study, it has been proven that the compound NAB inhibited the activation of LPS-induced RAW264.7 cells. It is clear that NAB increased the expression of HO-1 to reduce NO production. Also, inflammatory mediators, including NO, TNF-α, IL-1β, and IL-6, were inhibited by the pretreatment of NAB. More importantly, the study found that NAB has no inhibitory effect on COX-2, suggesting that it may be safer than NSAIDs. At the same time, our study found that the inhibitory effect of NAB on NO is similar to that of the positive control drug DEX, suggesting that NAB has great potential for future drug development. In conclusion, NAB may be a potential HO-1 inducer for the treatment of inflammatory diseases.

As described previously, the results suggested that NAB exerted its anti-inflammatory effects against LPS-induced inflammation via activating the Nrf2/HO-1 pathway (Figure 8). Also, the potential unique anti-inflammatory mechanism of NAB provides a new therapeutic solution for oxidative damage- and inflammation-related diseases. Although more experiments are needed in vivo and in vitro to verify the effect and mechanism of NAB in future research, this study helps to provide a potential treatment mechanism of *Nardostachys chinensis* and evidence for the use of this recently discovered natural compound in the treatment of diseases related to inflammation and oxidative stress.

## Figures and Tables

**Figure 1 molecules-24-02482-f001:**
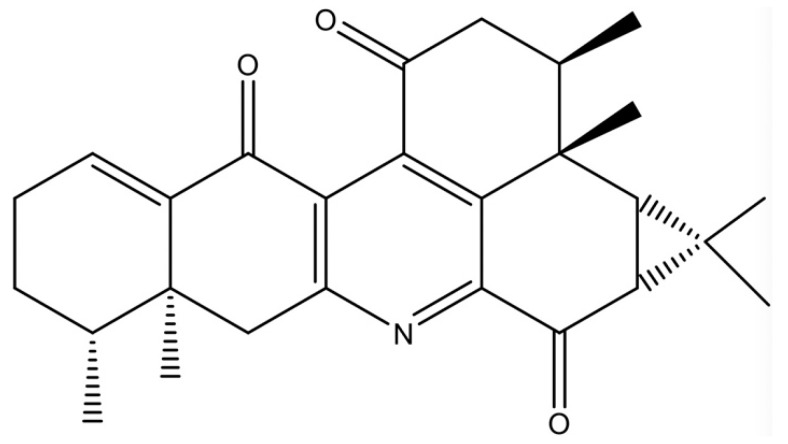
Chemical structure of nardochinoid B (NAB).

**Figure 2 molecules-24-02482-f002:**
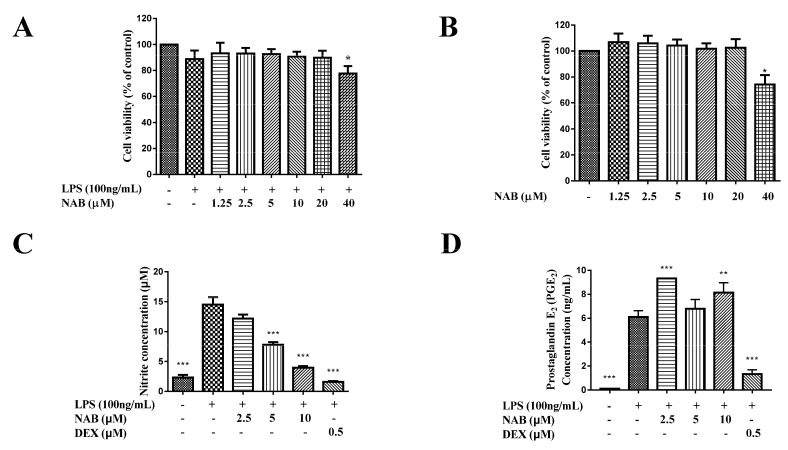
The effect of NAB on the release of nitric oxide (NO) and prostaglandin E_2_ (PGE_2_) in lipopolysaccharide (LPS)-induced RAW264.7 macrophages. (**A**) Cytotoxicity of NAB to LPS-stimulated RAW264.7 cells. (**B**) Cytotoxicity of NAB to normal RAW264.7 cells. Cells were treated with NAB at multiple concentrations (1.25, 2.5, 5, 10, 20, and 40 μM) for 1 h and then incubated with or without LPS stimulation (100 ng/mL) for 18 h. Cell viability was analyzed with the MTT method. (**C**) Effect of NAB on the production of NO by the LPS-stimulated RAW264.7 cells. (**D**) Effect of NAB on the production of PGE_2_ by the LPS-stimulated RAW264.7 cells. Cells were pretreated with NAB or the positive control drug (dexamethasone, DEX) for 1 h and then stimulated with or without LPS (100 ng/mL) for 18 h. Culture medium was collected, and the NO concentration was analyzed by the Griess reagent. The PGE_2_ concentration was measured by the ELISA method. The density ratio of the control group (blank control) in the cytotoxicity test was set to 1. In other tests, the variances were compared with the LPS group. Results are expressed as the mean ± SEM of three independent experiments. * *P* < 0.05, ** *P* < 0.01, and *** *P* < 0.001 vs. normal cells (**A**,**B**) or LPS-stimulated cells (**C**,**D**).

**Figure 3 molecules-24-02482-f003:**
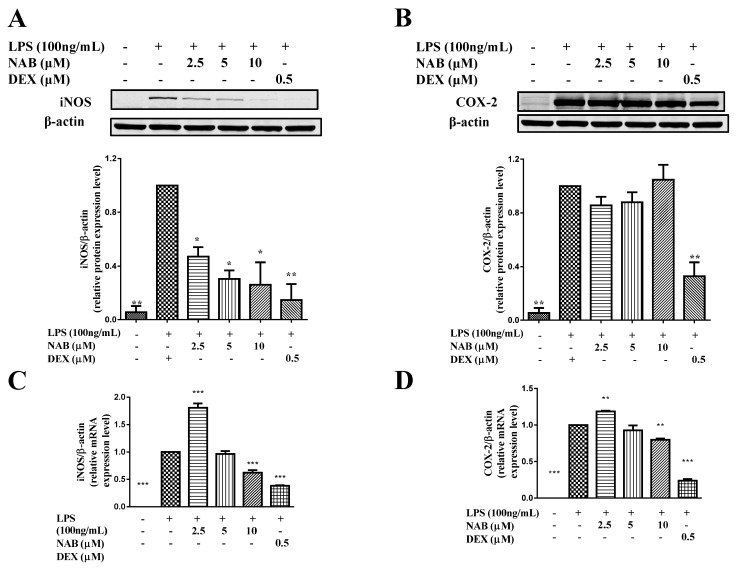
The effect of NAB on the expression levels of inducible nitric oxide synthase (iNOS) and cyclooxygenase-2 (COX-2) in LPS-induced RAW264.7 macrophages. (**A**,**B**) Effect of NAB on the expression levels of iNOS and COX-2 in LPS-stimulated RAW264.7 cells. Cells were pretreated with NAB or positive control drug (dexamethasone, DEX) for 1 h and then stimulated with LPS (100 ng/mL) for 18 h. The total protein of the cells was collected, and the expression levels of iNOS (**A**) and COX-2 (**B**) were analyzed with Western blotting; (**C**,**D**) Effect of NAB on the mRNA expression levels of iNOS and COX-2 in LPS-stimulated RAW264.7 cells. Cells were pretreated with NAB or positive control drug (dexamethasone, DEX) for 1 h and then stimulated with LPS (100 ng/mL) for 18 h. The total RNA was prepared, and the mRNA expression levels of iNOS (**C**) and COX-2 (**D**) were analyzed with qRT-PCR. The density ratio of the LPS group (model control) was set to 1. Results are expressed as the mean ± SEM of three independent experiments. * *P* < 0.05, ** *P* < 0.01, and *** *P* < 0.001 vs. LPS-stimulated cells.

**Figure 4 molecules-24-02482-f004:**
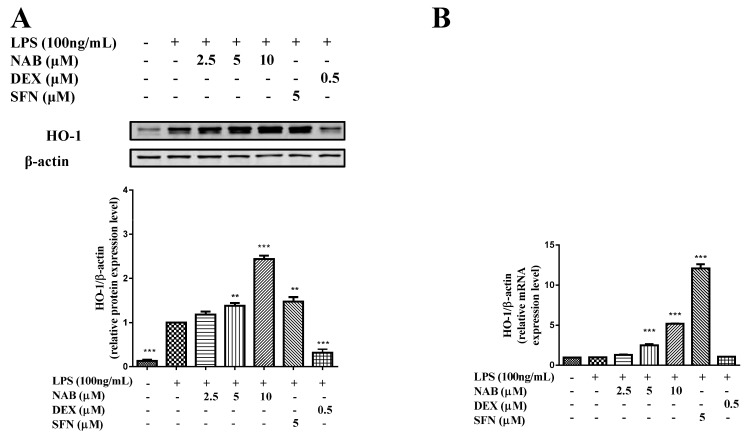
The effect of NAB on expression level of HO-1 in LPS-stimulated RAW264.7 cells. (**A**) Effect of NAB on HO-1 expression level in LPS-stimulated RAW264.7 cells. Cells were pretreated with NAB or positive control drug (dexamethasone, DEX) for 1 h and then stimulated with LPS (100 ng/mL) for 6 h. The total protein of the cells was prepared, and the expression level of HO-1 protein was measured by Western blotting. (**B**) Effect of NAB on the mRNA expression level in LPS-stimulated RAW264.7 cells. Cells were pretreated with NAB or positive control drug (dexamethasone, DEX, or sulforaphane, SFN) for 1 h and then stimulated with LPS (100 ng/mL) for 6 h. The total mRNA was prepared, and the expression level was evaluated by qRT-PCR. The density ratio of the LPS group (model control) was set to 1. Results are expressed as the mean ± SEM of three independent experiments. ** *P* < 0.01 and *** *P* < 0.001 vs. LPS-stimulated cells.

**Figure 5 molecules-24-02482-f005:**
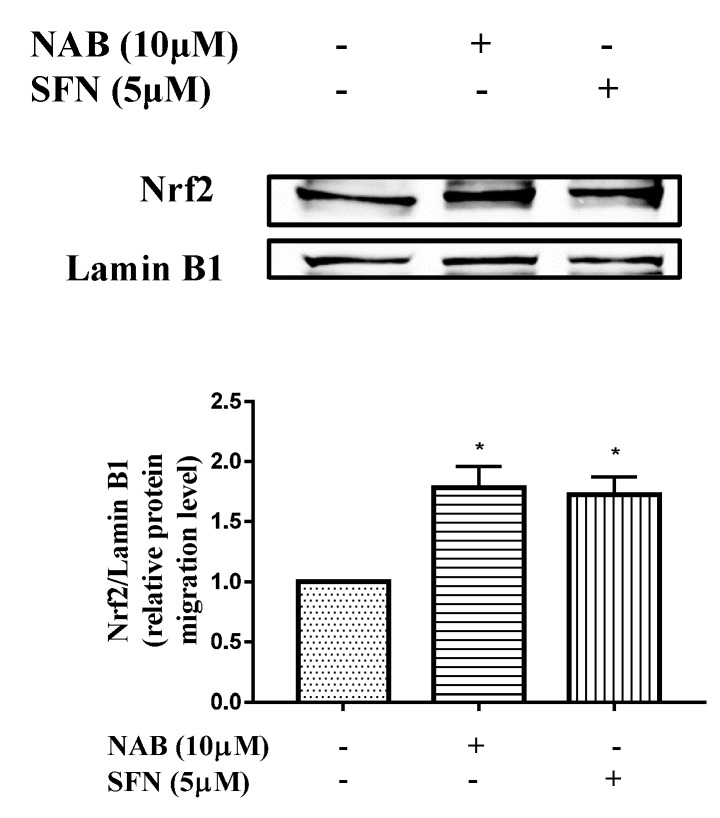
Effect of NAB on Nrf2 protein migration level in nucleoprotein of RAW264.7 macrophages. The RAW264.7 cells were treated with NAB or positive control drug (sulforaphane, SFN) for 6 h. The nuclear fraction was extracted, and the nuclear protein was measured by Western blotting. The density ratio of the CON group (normal control) was set to 1. Results are expressed as the mean ± SEM of three independent experiments. * *P* < 0.05 vs. normal cells.

**Figure 6 molecules-24-02482-f006:**
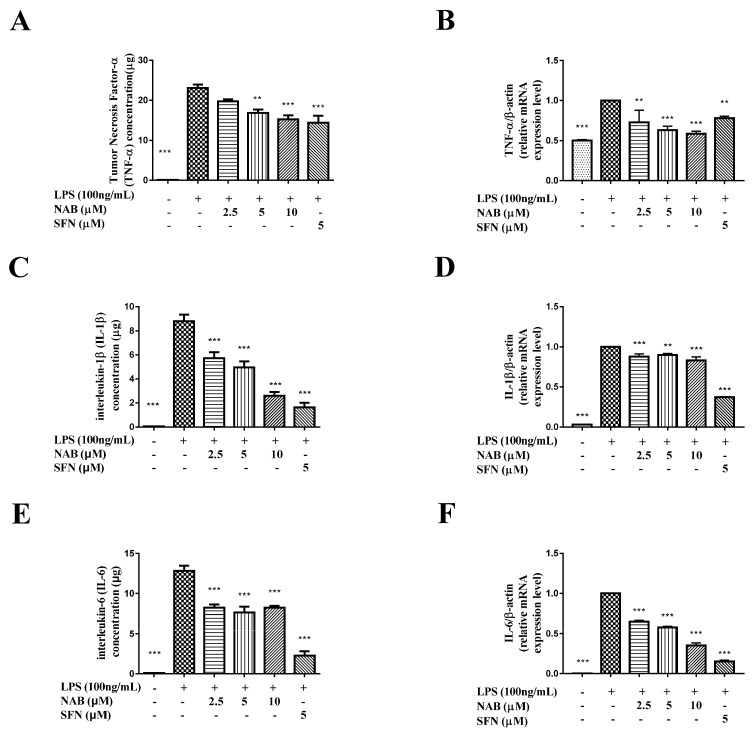
The effect of NAB on the production of TNF-α, IL-1β, and IL-6 in LPS-activated RAW264.7 cells. Effects of NAB on the expression level of TNF-α (**A**,**B**), IL-1β (**C**,**D**), and IL-6 (**E**,**F**) in LPS-stimulated RAW264.7 cells. Cells were pretreated with NAB or positive control drug (sulforaphane, SFN) for 1 h and then stimulated with LPS (100 ng/mL) for 18 h. Total mRNA was prepared, and the mRNA expression of TNF-α, IL-1β, and IL-6 was detected. Culture medium was collected, and ELISA was used to measure the expression level of TNF-α, IL-1β, and IL-6. In the qRT-PCR analysis, the density ratio of the LPS group (model control) was set to 1. In the ELISA analysis, the variances were compared with the LPS group. Results are expressed as mean ± SEM of three independent experiments. ** *P* < 0.01 and *** *P* < 0.001 vs. LPS-stimulated cells.

**Figure 7 molecules-24-02482-f007:**
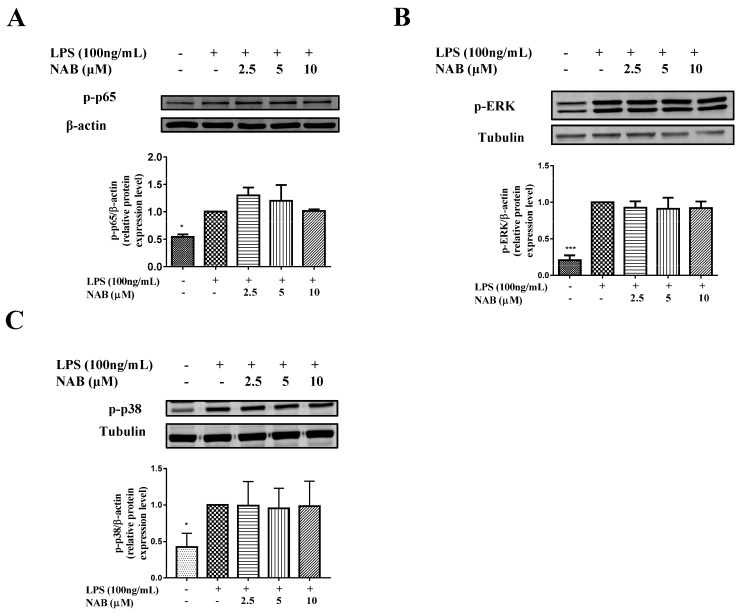
Effect of NAB on protein expression level of phospho-p65 (p-p65), phospho-extracellular regulated protein kinase (p-ERK), and phosphor-p38 (p-p38) in LPS-stimulated RAW264.7 cells. Cells were pretreated with NAB for 1 h and then stimulated with LPS (100 ng/mL) for 18 h. The total protein of the cells was collected, and the expression levels of p-p65 (**A**), p-ERK (**B**), and p-p38 (**C**) were analyzed with Western blotting. The density ratio of the LPS group (model control) was set to 1. Results are expressed as the mean ± SEM of three independent experiments. * *P* < 0.05 and *** *P* < 0.001 vs. LPS-stimulated cells.

**Figure 8 molecules-24-02482-f008:**
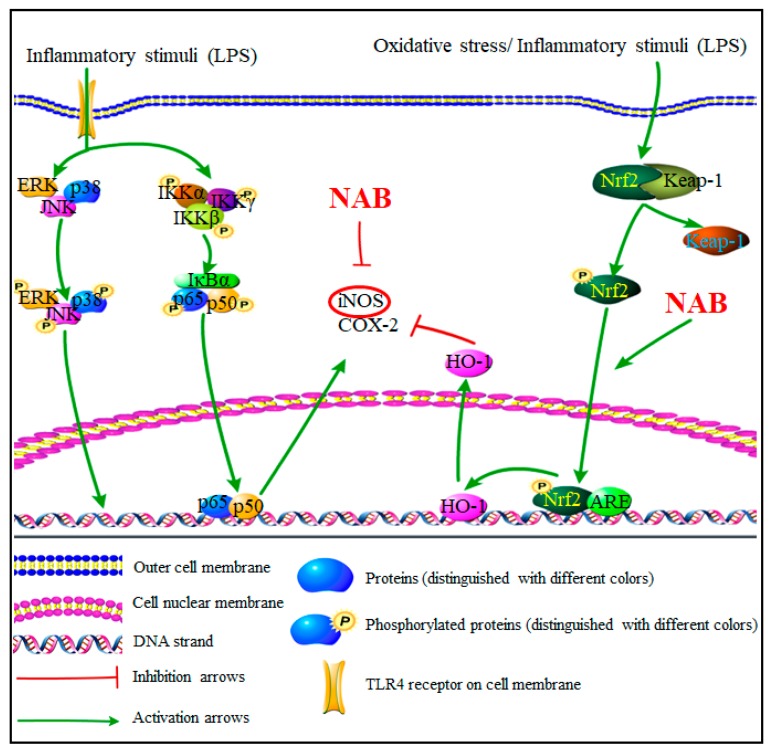
Proposed molecular mechanisms underlying the inhibitory effect of NAB on the LPS-activated RAW264.7 macrophages. NAB activates the Nrf2 pathway, leading to the high expression of HO-1, then contributing to the anti-inflammatory effects in LPS-activated RAW264.7 macrophages. Abbreviations in the figure: NAB, nardochinoid B; TLR4, Toll-like receptors; ERK, extracellular regulated protein kinase; JNK, c-Jun N-terminal kinase; HO-1, heme oxygenase (HO)-1; Nrf2, nuclear factor erythroid 2-related factor 2; Keap-1, Kelch-like ECH-associated protein-1; ARE, antioxidant response element; iNOS, inducible nitric oxide synthase; COX-2, cyclooxygenase-2; IKKα/β/γ, IκB kinase α/β/γ; IκBα, inhibitor of nuclear factor kappa-B kinase α.

**Table 1 molecules-24-02482-t001:** The primers used in this study.

Target Gene	Primer Sequences
β-actin_F	5′-CGGTTCCGATGCCCTGAGGCTCTT-3′
β-actin_R	5′-CGTCACACTTCATGATGGAATTGA-3′
iNOS_F	5′-CAGCACAGGAAATGTTTCAGC-3′
iNOS_R	5′-TAGCCAGCGTACCGGATGA-3′
COX-2_F	5′-TTTGGTCTGGTGCCTGGTC-3′
COX-2_R	5′-CTGCTGGTTTGGAATAGTTGCTC-3′
TNF-α_F	5′-TATGGCTCAGGGTCCAACTC-3′
TNF-α_R	5′-CTCCCTTTGCAGAACTCAGG-3′
IL-6_F	5′-GGTGACAACCACGGCCTTCCC-3′
IL-6_R	5′-AAGCCTCCGACTTGTGAAGTGGT-3′
HO-1_F	5′-CCCACCAAGTTCAAACAGCTC-3′
HO-1_R	5′-AGGAAGGCGGTCTTAGCCTC-3′
IL-1β_F	5′-TTGACGGACCCCAAAAGATG-3′
IL-1β_R	5′-AGAAGGTGCTCATGTCCTCA-3′

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
