# Peer review of "Nardochinoid B Inhibited the Activation of RAW264.7 Macrophages Stimulated by Lipopolysaccharide through Activating the Nrf2/HO-1 Pathway"

_molecules, 2019, doi:10.3390/molecules24132482_

Round 1

Reviewer 1 Report

Dear Authors,

Please find below my comments and some minor issues which are listed below.

The article focuses on the anti-inflammatory activity of Nardochinoid B (NAB) analysed in RAW264.7 macrophages stimulated by lipopolysaccharides (LPS) to immune response. Based on the results obtained, the Authors have proposed molecular mechanisms underlying the inhibitory effect of NAB on the LPS-activated RAW264.7 macrophages. They found that NAB mitigates LPS-induced inflammation by down-regulation the mRNA expression of inducible nitric oxide synthase (iNOS) and reducing NO generation without influence on cyclooxygenase-2 expression and prostaglandin E2 production. They proved that NAB activates Nrf2 pathway lead to the high expression of HO-1, thus contributes to the anti-inflammatory effects in LPS-activated RAW264.7 macrophages via activating the Nrf2/HO-1 pathway. As a result, the production of pro-inflammatory cytokines: TNF-α, IL-1β and IL-6 is significantly suppressed.  I regret that the Authors did not analyse the expression of the anti-inflammatory IL-10 cytokine, which is involved in OH-1 pathway.

The experiments were well designed. All controls (both negative and positive) needed to verify the anti-inflammatory NAB potential were included in the study.

From a general point of view, I have an only comment related to the manuscript organisation. Because the result section contains several elements of discussion, such as introduction to the results discussed, description of signalling pathways, and citations, I would suggest combining Results and Discussion sections into one. Such an arrangement would eliminate the repetition and reduce the manuscript volume.

Specific comments are as follows:

- There is no information on the statistical analysis used in the studies. Please, complete the description in the Materials and Methods section.

- line 146 – “…after stimulation with LPS…” instead of “…after stimulated with LPS…”

- Figure 4A - probably a mistake in the description of X-Axis for the second bar (“-“ instead of “+” for DEX). 

- Figure 4A - The results obtained for HO-1 protein expression in LPS-activated RAW264.7  macrophages treated with sulforaphane (SFN) are not shown in the figure.

- Figure 7 - The figure does not contain error bars (standard deviation), and no statistically significant differences are marked.

- The description of the results includes information that is part of the discussion. I wonder if maybe the results and discussion sections should be combined. I leave this issue to the Authors and Editors for consideration.

Author Response

2rd June 2019

Editorial Office

Molecules

Dear Editor,

Ref: Nardochinoid B inhibited the activation of RAW264.7 macrophages stimulated by lipopolysaccharide through activating Nrf2/HO-1 pathway

Thank you and reviewers for providing us the valuable comments and suggestions on our manuscript. According to these comments and suggestions, we have revised the manuscript and attach here the revision of our manuscript for your consideration of publication in Molecules. All the modified words or sentences have been incorporated in the revised manuscript and these modifications have been marked in red for ease of reference.

Reviewer #1:

Point 1: There is no information on the statistical analysis used in the studies. Please, complete the description in the Materials and Methods section.

Response 1: Thank you for this suggestion. We have added the related description in Materials and methods in red and listed below here.

4.7 Data analysis

All the data was presented as mean ± SEM for 3 individual experiments. Difference were analyzed by one-way ANOVA by using GraphPad Prism 7. In all cases, a level of P< 0.05 was considered statistically significant. One-way ANOVA was used in all statistical tests, the post-hoc analysis method was Tukey’s multiple comparison test.

Point 2: line 146 – “…after stimulation with LPS…” instead of “…after stimulated with LPS…

Response 2: Thank you for this valuable suggestion. We have altered the words in the manuscript in red and listed below here.

The results (Figure 4) show that the expression level of HO-1 was increased after stimulation with LPS (100ng/ml) for 6 h.

Point 3: Figure 4A - probably a mistake in the description of X-Axis for the second bar (“- “instead of “+” for DEX).

Response 3: Thank you for the comment. We have replaced the figure in the manuscript in Figure 4 and the cover letter in Word version we submitted.

Point 4: Figure 4A - The results obtained for HO-1 protein expression in LPS-activated RAW264.7 macrophages treated with sulforaphane (SFN) are not shown in the figure.

Response 4: Thank you for the comment. We have replaced the figure in the manuscript in Figure 4 and the cover letter in Word version we submitted.

Point 5: The figure does not contain error bars (standard deviation), and no statistically significant differences are marked.

Response 5: Thank you for this comment. We have replaced the figure (Figure 7) in the manuscript and the cover letter in Word version we submitted.

Point 6: The description of the results includes information that is part of the discussion. I wonder if maybe the results and discussion sections should be combined. I leave this issue to the Authors and Editors for consideration.

Response 6: Thank you for your valuable suggestion. We have rewritten the Result and the Discussion part in red in the manuscript and these parts also listed below here.

2. Result

2.1. Anti-inflammatory activities of NAB on LPS-activated RAW264.7 macrophages

2.1.1. NAB reduced the release of NO in LPS-stimulated RAW264.7 macrophages

The results from the MTT assay (Figure 2) show that NAB had no significant cytotoxicity to LPS-stimulated RAW264.7 cells at the concentrations lower than 20μM (Figure 2A-B). The nitrite level (evaluated through the stable oxidized product of NO) and the production of PGE2 in the culture medium of the RAW264.7 cells were significantly increased (P<0.01) after 18h LPS stimulation. The pretreatment with NAB has markedly decreased the LPS-induced NO production in a concentration-dependent manner (Figure 2C), while it did not inhibit the production of PGE2 (Figure 2D). Dexamethasone (shorted as DEX), was selected as positive drug. The results show that the positive drug DEX markedly reduced the production of both NO and PGE2 in LPS-stimulated RAW264.7 cells (Figure 2C-D).

2.1.2 NAB inhibited the expression of iNOS rather than COX-2 in LPS-stimulated RAW264.7 macrophages

The mRNA and protein expression of iNOS and COX-2 in the cells were significantly increased after stimulated with LPS (100 ng/ml) for 18 h (Figure 3). NAB markedly down-regulated the protein expression level of iNOS in the LPS-stimulated RAW264.7 cells in a concentration-dependent manner (Figure 3A) and decreased the mRNA expression of iNOS at the concentration of 10μM (Figure 3C). However, NAB did not significantly down-regulate the mRNA and protein expression levels of COX-2 at the same condition (Figure 3B&D).

2.2 Potential anti-inflammatory mechanisms of NAB on LPS-induced RAW 264.7 macrophages

2.2.1 NAB increased the mRNA and protein expression levels of HO-1 in LPS-stimulated RAW264.7 cells.

The results (Figure 4) show that the expression level of HO-1 was increased after stimulation with LPS (100ng/ml) for 6 h. Except for DEX, sulforaphane (short as SFN) was selected as the positive drug as well. NAB significantly increased the mRNA (Figure 4A) and protein (Figure 4B) expression levels of HO-1 in LPS-stimulated RAW264.7 cells.

2.2.2 NAB promoted Nrf2 protein translocated into nucleus in RAW 264.7 macrophages.

The pretreatment of NAB promoted Nrf2 protein entering into the nucleus in RAW264.7 cells, similar to the effect of SFN, the positive control in the experiment.

2.2.3 NAB suppressed the productions of TNF-α, IL-1β and IL-6.

The results show that the stimulation of LPS to RAW264.7 cells increased the expression levels of TNF-α (Figure 6A), IL-1β (Figure 6C) and IL-6 (Figure 6E) in the culture medium. Also, the mRNA expression levels of TNF-α (Figure 6B), IL-1β (Figure 6D) and IL-6 (Figure 6F) were induced by LPS as well. The treatment of NAB down-regulated the expression levels of TNF-α (Figure 6A&B), IL-1β (Figure 6C&D) and IL-6 (Figure6E&F). The positive drug sulforaphane also significantly inhibited the expression level of these inflammatory mediators (Figure6).

2.2.4 NAB failed to inhibit the activation of NF-κB and MAPK pathway in LPS-stimulated RAW264.7 cells

As shown in Figure 7, the stimulation of LPS to RAW264.7 cells increased the expression levels of p-p65, p-p38 and p-ERK. However, NAB failed to inhibit the increased expression levels of p-p65 (Figure 7A), p-ERK (Figure 7B) and p-p38 (Figure 7C).

3. Discussion

As described before, the macrophages play an important role in inflammation since it could release kinds of cytokines to ignite inflammatory reactions [16]. The LPS-stimulated RAW264.7 macrophages is a kind of classical inflammatory cell model and widely used in evaluating the anti-inflammatory effect and mechanisms of many natural products derived from the Chinese medicines [24]. Therefore, we chose the LPS-stimulated RAW264.7 cells as the cell model in this study. Dexamethasone (DEX) and Sulforaphane (SFN) were chosen as the positive drugs in this study. DEX is a kind of classic anti-inflammatory drug that is widely used in clinic [25]. It is a kind of steroidal anti-inflammatory drug that has been widely used in clinic to treat rheumatoid arthritic knees [26], pneumonia [27] and bronchiolitis [28]. SFN is a kind of drug that has been confirmed as the Nrf2 activator. It is a natural isothiocyanate and it has been proved that it could suppress LPS-induced inflammation in mouse peritoneal macrophages through activating the Nrf2 pathway and up-regulating the HO-1 expression [29]. Moreover, SFN inhibited the expression of some inflammatory mediators, including TNF-α, IL-1β and IL-6 [30] through activating Nrf2 pathway. So DEX was chosen as the positive drug in the study to evaluate the anti-inflammatory activity of NAB and SFN was chosen as the positive control in the study to evaluate the Nrf2 pathway related mechanism of NAB.

In this study, we firstly evaluated the cytotoxicity of NAB and found out that NAB had no significant cytotoxicity of LPS-stimulated RAW 264.7 cells at the concentration lower than 20 μM (Figure 2A-B). Thus, we have selected the concentration of NAB ranging from 2.5 μM to 10 μM to examine the anti-inflammatory activity of NAB in the LPS-stimulated RAW 264.7 cells. Then, followed by evaluated the effect of NAB on the production of NO and PGE2 in the LPS-induced RAW 264.7 cells, we have examined the effect of NAB on the expression of iNOS and COX-2 in LPS-stimulated RAW 264.7 macrophages since iNOS and COX-2 are the enzymes responsible for the production of NO and PGE2 respectively. After that, the expression level of HO-1 in LPS-stimulated RAW 264.7 cells was detected with the treatment of NAB because the HO-1 is one of the regulating factors of the expression of iNOS. Since the translocation of Nrf2 protein into the cell nucleus mediates the expression of HO-1, the migration level of Nrf2 protein in the RAW 264.7 macrophages was evaluated. Moreover, since the macrophages release cytokines (e.g. TNF-α, IL-1β and IL-6) [] to promote and encourage the development and progression of inflammation in vivo, these inflammatory mediators were detected in this study.

NO and PGE2 are two of the most important inflammatory mediators that participate in the inflammatory processes. The inflammation and the exposure of tissue cells to bacterial products such as LPS, lipoteichoic acid (TLA) peptidoglycan and bacterial DNA or whole bacteria will induce the high expression of iNOS and then enhance the production of NO. In these situations, the NO forms peroxynitrite, acts as a cytotoxic molecule, resists invading microorganisms and acts as a killer [31]. However, it has been reported that in aseptic inflammation, the iNOS expression and NO formation would also be induced in human macrophages, for example, in rheumatoid arthritis and osteoarthritis [32]. In these bacterial-free inflammatory processes, the synthesis of NO can be an important factor helps maintaining the inflammatory and osteolytic processes [20]. PGE2 mediates the increasing of arterial dilation and microvascular permeability [33]. This action will cause blood to flow into the inflamed tissue and then causes redness and edema [34]. COX-2 belongs to the regulatory enzyme for the production of PGE2, it also regulates the synthesis of prostaglandin I2 (PGI2, also called as prostacyclin) and thromboxaneA2 (TXA2) [35]. It is known that TXA2 is the major cyclooxygenase product in the platelet. It is also a potent vasoconstrictor and can stimulate the aggregation of platelet in vitro. PGI2 is produced and synthesized in vascular endothelial cells. It is a vasodilator and inhibitor of platelet [35]. It has been proven that the inhibition of COX-2 may break the balance between PGI2 and TXA2 and then causes cardiovascular risks [36]. Fortunately, in this study, the results show that NAB only targets on inhibiting NO and iNOS and does not affect the expression of COX-2 (Figure 2C&D and Figure 3). Thus, NAB may have low cardiovascular side effects compared with DEX. However, the results showed that the protein expression level of iNOS was inhibited by the concentration of 2.5μM NAB while the mRNA expression level was not, it was considered that NAB may affect the translate process of iNOS from gene to protein.

TNF-α, IL-1β and IL-6 are belong to inflammatory cytokines and are able to involve in inflammatory processes, too [37]. In this research, all these inflammation cytokines and regulatory enzymes (iNOS and COX-2) were upregulated by the LPS stimulation (Figure 3, 6). Then, the increases of these inflammatory mediators were significantly inhibited by NAB (Figure 6). More importantly, the inhibitory effect of NAB on the mRNA expression level of TNF-α is better than that of SFN, meaning that NAB has obvious anti-inflammatory activity in LPS-stimulated RAW264.7 macrophage cells.

Usually, the inflammatory processes accompany with the activation of NF-κB pathway, which also promotes the expression of inflammatory mediators in macrophages [38]. Previous study has showed that the production of inflammatory cytokines is related to the LPS-induced activation of NF-κB pathway [39]. Mitogen-activated protein kinases (MAPK) pathway also plays an critical role in inflammatory responses [40]. The activation of both NF-κB and MAPK signaling pathways involve in the development of inflammation [24]. Therefore, under normal circumstances, inhibiting NF-κB and MAPK signaling pathways are considered as effective ways to against inflammatory reaction. The activation of NF-κB resulted in the phosphorylation of IκBα, IKKα/β, p65, and then leading to the transcription of inflammatory genes and the expression of inflammatory proteins [41]. The activation of MAPK pathway results in the phosphorylation of p38, JNK and ERK [42], which may promote pro-inflammatory cytokines production [43]. Some reports showed that the deactivation of NF-κB and MAPK pathway in RAW 264.7 cells leads to the inhibition of LPS-induced NO, PGE2, iNOS, COX-2, TNF-α and IL-6 productions [24,44]. Other studies have reported that the extracts of N. chinensis inhibited p38 MAPK pathway to inhibit the expression of inflammatory mediators [45]. To study the anti-inflammatory mechanism of NAB, we first investigated the effect of NAB on the activations of NF-κB and MAPK pathways in LPS-stimulated RAW264.7 cells. However, the results showed that NAB didn’t inhibit the activations of NF-κB and MAPK pathways (Figure 7), so NAB may not act on NF-κB and MAPK pathways to exert its anti-inflammatory effects.

The activation of Nrf2 pathway is another possible way to prevent LPS-induced transcriptional upregulation of pro-inflammatory cytokines, including TNF-α, IL-1β and IL-6 [46]. These inflammatory cytokines were decreased by the Nrf2-dependent anti-oxidant genes (HO-1 and NQO-1). In Nrf2-knockout mice, the mRNA and protein levels of COX-2, iNOS, IL-6, and TNF-α increased [47] and the anti-inflammatory effect also disappeared [48]. Since the current result showed that NAB inhibited TNF-α, IL-1β and IL-6 obviously (Figure 6), so it was hypothesized that NAB may activate the Nrf2 pathway to exert its anti-inflammatory effect.

In this study, it was found that NAB inhibited the expression of inflammatory protein iNOS (Figure 3A&C) and inflammatory cytokines, including NO, TNF-α and IL-6 (Figure 2C and Figure6) accompanied with the increase of antioxidant protein HO-1 (Figure 4). More importantly, the study found that NAB had no inhibitory effect on COX-2 (Figs. 3B&D) and PGE2 (Figure 2D), suggesting that NAB has potential to be developed as selective iNOS/NO inhibitor, a kind of anti-inflammatory drugs helps to reduce the airway inflammatory responses like the 1400W [49] and relieve the pain caused by mechanical damage like the AR-C102222 [50]. More than that, since NAB did not affect the expressions of COX-2 and PGE2, it is safer than NSAIDs, which inhibit PGE2 to exert its anti-inflammatory effect through inhibiting the COX-2 expression. At the same time, our study found that the inhibitory effect of NAB on NO is approximate to positive drug DEX, suggesting that NAB has great development value in future study.

The oxidative or nitrosative stress, cytokines and other mediators may cause the cells to over produce HO-1 to affect the cell protection[51,52]. The induction of HO-1 reduces the production of inflammatory mediators and modulates the inflammatory process [53]. HO-1 can be rapidly induced by various oxidative-inducing agents, including LPS [51]. The current results also show that LPS increased the level of HO-1 slightly, but compared with LPS group, NAB further increased the level of HO-1 protein dramatically (Figure 4). The NO production induced by LPS was inhibited by the high expression of HO-1 [54]. In this study, NAB reduced NO production while increasing HO-1 expression (Figure4), the current results are consistent with that the high expression of HO-1 can inhibit LPS-induced NO production [54].

Another factor that relates with the expression of HO-1 is the expression of interleukin (IL)-10. IL-10 induces the phosphorylation Janus Kinase (Jak) 1 and the activation of signal transducer and activator of transcription (STAT)-1 and STAT-3[55]. Also, IL-10 activates phosphatidylinositol-3 kinase (PI3K), which involves in the proliferative effects of the cytokine [56]. It has been proved that the IL-10 can induce the expression of HO-1 [57]. Also, it is reported that the IL-10-induced activation of STAT-3 and PI3K were associated with the expression of HO-1 [58]. Thus, we will do some further studies of NAB on the expression of IL-10 and the associated production like STAT-1 and PI3K.

Taken together, the results suggested that the activation the Nrf2/HO-1 pathway is the potential mechanism of NAB to exert its anti-inflammatory effects against LPS-activated inflammation (Figure 8).

Point 7: I regret that the Authors did not analyze the expression of the anti-inflammatory IL-10 cytokine, which is involved in HO-1 pathway.

Response 7: Thank you for this comment and suggestion. We agree that IL-10 cytokine also participates in the regulation of the expression of HO-1. The IL-10 induces the expression of HO-1 via activating the signal transducer and activator of transcription (STAT)-3. However, the expression of HO-1 also required the activation of phosphatidylinositol-3 kinase (PI3K) pathway, which can be activated by IL-10 but is not related to the anti-inflammatory effect of IL-10. Moreover, not only IL-10 but also IL-6 can induce the activation of STAT-3 to upregulate the expression of HO-1. Thus, we did not analyze the effect of NAB on the expression of IL-10 in the study. But we are appreciated that you suggested to evaluate the effect of NAB on the expression of IL-10. And we will analyze the expression of IL-10 in the NAB-treated cells and study the relationship between them in the further study. Nevertheless, we have added the discussion of IL-10 in the manuscript in red and it also shows below here:

Another factor that relates with the expression of HO-1 is the expression of interleukin (IL)-10. IL-10 induces the phosphorylation Janus Kinase (Jak) 1 and the activation of signal transducer and activator of transcription (STAT)-1 and STAT-3[55]. Also, IL-10 activates phosphatidylinositol-3 kinase (PI3K), which involves in the proliferative effects of the cytokine [56]. It has been proved that the IL-10 can induce the expression of HO-1 [57]. Also, it is reported that the IL-10-induced activation of STAT-3 and PI3K were associated with the expression of HO-1 [58]. Thus, we will do some further studies of NAB on the expression of IL-10 and the associated production like STAT-1 and PI3K.

I hope the above responses and the revised manuscript can satisfy the reviewers’ questions and comments. Should you have any queries, please do not hesitate to contact me at (+853) 8897 2458. Thank you again for consideration of our work. I look forward hearing from you.

Yours sincerely,

Hua Zhou, PhD

Professor

The State Key Laboratory of Quality Research in Chinese Medicine,

Macau Institute for Applied Research in Medicine and Health,

Macau University of Science and Technology

Taipa, Macau

huazhou2009@gmail.com

+853 88972458

Reviewer 2 Report

Yao et al. investigated the anti-inflammatory effects of Nardochinoid B (NAB), a compound that is stated to be known from traditional Chinese medicine. The authors performed in vitro analysis using a murine macrophage cell line to investigate the anti-inflammatory properties and the underlying mechanisms with a focus on inducible nitric oxide synthase (iNOS) and cyclooxygenase-2 (COX-2).

The topic of this manuscript is interesting. However, I have major concerns and suggestions.

Major comments:

Figure 2 – 7: In the manuscript the results are based on “three independent experiments”. The n-number of 3 is too low to perform adequate statistical analysis. Furthermore, the statement of the used statistical test is missing.

Figure 5: The quality of the blot is very low. Therefore, I question the densitometry analysis plus the n-number is too low.

Figure 7: The author should state the n-number of the experiments. No statistical analysis was performed.

Figure 7B: Why is there a double band appearing?

Line 183 - 184: The figure references are wrong.

Line 194 - 195: Where did the authors analyze a density ratio in this figure?

Figure 8: The figure legend is incomplete. The definition of all abbreviations is missing. Furthermore, some "structures" are not labeled. For example, what kind of receptor/channel recognizes LPS? How can extracellular oxidative stress/Inflammatory stimuli (LPS) activate intracellular Nrf2? Is the blue line resembling the outer cell membrane?

Line 309 – 310: What kind of solvent is used for NAB?

Line 360 - 364: The references of the primary antibodies are missing.

Minor comments:

Line 75-76: Please rephrase the sentence.

Line 146: 100 ng/ml

Line 166: SFN abbreviation was defined before.

Line 176: Please rephrase this sentence. Macrophages are not only activated following the occurrence of inflammation. They encourage inflammation by releasing cytokines and so on.

Author Response

2rd June 2019

Editorial Office

Molecules

Dear Editor,

Ref: Nardochinoid B inhibited the activation of RAW264.7 macrophages stimulated by lipopolysaccharide through activating Nrf2/HO-1 pathway

Thank you and reviewers for providing us the valuable comments and suggestions on our manuscript. According to these comments and suggestions, we have revised the manuscript and attach here the revision of our manuscript for your consideration of publication in Molecules. All the modified words or sentences have been incorporated in the revised manuscript and these modifications have been marked in red for ease of reference.

Reviewer #2:

Point 1: Figure 2 – 7: In the manuscript the results are based on “three independent experiments”. The n-number of 3 is too low to perform adequate statistical analysis. Furthermore, the statement of the used statistical test is missing.

Response 1: Thank you for your comment and suggestion. All of the results in our study were analyzed by one-way ANOVA by using GraphPad Prism 7. We have added the description of data analysis in Materials and methods in red and listed below here. Moreover, all of the results in our study were evaluated and analyzed from three independent samples, which were gotten from three independent experiments. We believe that the “three independent experiments” in vitro is enough to prove our statements.

Note: The description of data analysis is listed below here:

4.7 Data analysis

All the data was presented as mean ± SEM for 3 individual experiments. Difference were analyzed by one-way ANOVA by using GraphPad Prism 7. In all cases, a level of P< 0.05 was considered statistically significant. One-way ANOVA was used in all statistical tests, the post-hoc analysis method was Tukey’s multiple comparison test.

Point 2: Figure 5: The quality of the blot is very low. Therefore, I question the densitometry analysis plus the n-number is too low.

Response 2: Thank you for this comment. We have used new Western blot images in Figure 5 in the manuscript and the cover letter in Word version we submitted.

Point 3: Figure 7: The author should state the n-number of the experiments. No statistical analysis was performed.

Response 3: Thank you for your valuable comment. We have replaced the figure (Figure 7) in the manuscript and the cover letter in Word version we submitted.

Point 4: Figure 7B: Why is there a double band appearing?

Response 4: Thank you for this comment. Figure 7B shows the effect of NAB on the expression of p-ERK protein. The p-ERK protein includes p-ERK1 and p-ERK2 and the protein molecular weight of them are 42 kDs and 44 kDs. Thus, the figure show a double band there.

Point 5: Line 183 - 184: The figure references are wrong.

Response 5: Thank you for the comment. We have rewritten the figure references and they are listed in red in the manuscript and below here:

The treatment of NAB down-regulated the expression levels of TNF-α (Figure 6A&B), IL-1β (Figure 6C&D) and IL-6 (Figure 6E&F).

Point 6: Line 194 - 195: Where did the authors analyze a density ratio in this figure?

Response 6: Thank you for this valuable comment. We have added the description of data analysis in the manuscript (Response 1) and modified the description of the figure legends in Figure 7 and they are listed in red in the manuscript and also below here:

4.7 Data analysis

All the data was presented as mean ± SEM for 3 individual experiments. Difference were analyzed by one-way ANOVA by using GraphPad Prism 7. In all cases, a level of P< 0.05 was considered statistically significant. One-way ANOVA was used in all statistical tests, the post-hoc analysis method was Tukey’s multiple comparison test.

Effects of NAB on the expression level of TNF-α (A, B), IL-1β (C, D) and IL-6 (E, F) in LPS-stimulated RAW264.7 cells. Cells were pretreated with NAB or positive drug (Sulforaphane, short as SFN) for 1h, and then stimulated with LPS (100 ng/ml) for 18 h. Total mRNA was prepared, the mRNA expressions of TNF-α, IL-1β and IL-6 was detected. Culture medium was collected, and ELISA was used to measure the expression level of TNF-α, IL-1β and IL-6. In the q-RT-PCR analysis, the density ratio of LPS group (model control) was set to 1. In the ELISA analysis, the variances were compared with LPS group. Results are expressed as mean±SEM of three independent experiments. ** P<0.01, *** P<0.001, vs. LPS-stimulated cells.

Point 7: Figure 8: The figure legend is incomplete. The definition of all abbreviations is missing. Furthermore, some "structures" are not labeled. For example, what kind of receptor/channel recognizes LPS? How can extracellular oxidative stress/Inflammatory stimuli (LPS) activate intracellular Nrf2? Is the blue line resembling the outer cell membrane?

Response 7: Thank you for your valuable comment. We have added the abbreviations and labeled the structures. Also, we have improved the figure legend in Figure 8. The new figure legend is listed in red in the manuscript with the completed figure 8 and please check the figure in the cover letter in Word version.

.

Figure 8. Proposed molecular mechanisms underlying the inhibitory effect of NAB on the LPS-activated RAW264.7 macrophages.

NAB activated Nrf2 pathway lead to the high expression of HO-1, then contributed to the anti-inflammatory effects in LPS-activated RAW264.7 macrophages.

Abbreviations in the figure: NAB, Nardochinoid B; TLR4, Toll-like receptors; ERK, extracellular regulated protein kinases; JNK, c-Jun N-terminal kinase; HO-1, heme oxygenase (HO)-1; Nrf2, nuclear factor erythroid 2-related factor 2; Keap-1, kelch-like ECH-associated protein-1; ARE, antioxidant response element; iNOS, inducible nitric oxide synthase; COX-2, cyclooxygenase-2; IKKα/β/γ, IκB kinase α/β/γ; IκBα, inhibitor of nuclear factor kappa-B kinase α.

Point 8: Line 309 – 310: What kind of solvent is used for NAB?

Response 8: Thank you for the comment. The solvent used for NAB was Dimethyl Sulfoxide (DMSO). We have added the description of solvent in the 4.1 Materials in red in the manuscript and it is also listed below here:

Dimethyl sulfoxide (DMSO) (Sigma, Cat. No. D2625) was used to dissolved the NAB powder and the concentration of mother liquid was 30 mM.

Point 9: Line 360 - 364: The references of the primary antibodies are missing.

Response 9: Thank you for your comment. We have added the references of the primary antibodies in the 4.1 Materials in red in the manuscript and it is also listed below here:

Antibodies to iNOS [59], COX-2 [60], heme oxygenase (HO)-1 [60], Nrf2 [10], p-p65[10], p-p38 [10] and phospho-extracellular regulated protein kinases (p-ERK) [10] were obtained from Cell Signaling Technology (Boston, MA, USA). Antibodies to β-actin and Laminin B1 [10] were from Santa Cruz Biotechnology (Santa Cruz, CA, USA). Antibody to α- Tubulin [10] was from Sigma Chemical Co. (St. Louis, MO, USA).

Point 10: Line 75-76: Please rephrase the sentence.

(Macrophages plays an important role in the innate immune response. It promotes cell protection and repair processes [16].)

Response 10: Thank you for this comment. We have rephrased the sentence and listed it below here in red:

Macrophages plays important roles in the innate immune response. It protects cell from injury induced by exogenous factors like bacterial and virus and endogenous factors like other damaged cells. Also, macrophages promote the repair processes of tissue injury.

Point 11: Line 146: 100 ng/ml

Response 11: Thank you for the comment. We have changed the words in (100 ng/ml)

Point 12: Line 166: SFN abbreviation was defined before.

Response 12: Thank you for this comment. We have modified the description of 2. Results and 3. Discussion parts in the manuscript and the definition of NAB is deleted in the part.

Point 13: Line 176: Please rephrase this sentence. Macrophages are not only activated following the occurrence of inflammation. They encourage inflammation by releasing cytokines and so on.

Response 13: Thank you for this valuable comment. We have modified the description of 2. Results and 3. Discussion parts in the manuscript and this sentence is rephrased in 3. Discussion part and listed in red below here:

Moreover, since the macrophages release cytokines (e.g. TNF-α, IL-1β and IL-6) [31]to promote and encourage the development and progression of inflammation in vivo, these inflammatory mediators were detected in this study.

I hope the above responses and the revised manuscript can satisfy the reviewers’ questions and comments. Should you have any queries, please do not hesitate to contact me at (+853) 8897 2458. Thank you again for consideration of our work. I look forward hearing from you.

Yours sincerely,

Hua Zhou, PhD

Professor

The State Key Laboratory of Quality Research in Chinese Medicine,

Macau Institute for Applied Research in Medicine and Health,

Macau University of Science and Technology

Taipa, Macau

huazhou2009@gmail.com

+853 88972458

Reviewer 3 Report

The authors reported the anti-inflammatory activity of nardochinoid B(NAB) and its underlying mechanism using the LPS-stimulated RAW264.7 cells. The authors emphasize that NAB does not inhibit COX2 and that its anti-inflammatory mechanism depends on Nrf2/HO-1. Several issues need to be clarified to be published in this journal.

As shown in the reference 25, activation of Nrf2 leads to the suppression of iNOS, TNFa, IL-6 and COX2. If NAB activate Nrf2, then how the authors explain the effect of NAB on COX2.

The authors used dexamethasone as a positive drug for NO and PGE2. Although dexamethasone inhibit COX2, it does not belong to NSAID. The anti-inflammatory activity of Dex goes beyond COX inhibitors.

In Introduction, the authors should give a brief detail on NAB and the Nrf/HO pathway. Most of all, the relevance of Nrf to inflammation needs to be described.

It would be better to compare NAB with Dex and SFN together. It is confusing to use these reference compounds in different assays.

The authors only checked p-p65, p-p38 and p-ERK. P-JNK and IkBa degradation need to be verified.

At least cytokine release should be evaluated using peritoneal macrophages.

Figure 2D shows that NAB does not inhibit PGE2. Remove the following COX2 gene and proteins in Figure 2  

Minor ones

The qualities of several Western blot bands are poor. Replace figure 5 with the better one.

Lines 75-78 need to be relocated in the first and second paragraphs.

Check Figure6A. It contains TNFa in microM

Author Response

2rd June 2019

Editorial Office

Molecules

Dear Editor,

Ref: Nardochinoid B inhibited the activation of RAW264.7 macrophages stimulated by lipopolysaccharide through activating Nrf2/HO-1 pathway

Thank you and reviewers for providing us the valuable comments and suggestions on our manuscript. According to these comments and suggestions, we have revised the manuscript and attach here the revision of our manuscript for your consideration of publication in Molecules. All the modified words or sentences have been incorporated in the revised manuscript and these modifications have been marked in red for ease of reference.

Reviewer #3:

Point 1: As shown in the reference 25, activation of Nrf2 leads to the suppression of iNOS, TNF-α, IL-6 and COX2. If NAB activate Nrf2, then how the authors explain the effect of NAB on COX2.

Response 1: Thank you for this comment. It is true that the activation of Nrf2 leads to the suppression on COX-2. However, the activation of Nrf2 is not the only signaling pathway that regulates the expression of COX-2. Also, the “expression of COX-2” which can be measured is the COX-2 that “exist” in the cells. But the “exist” of COX-2 associates with not only the “expression” but also the “degration” of COX-2. The effect of NAB on the degration of COX-2 and other pathways that relates to the expression of COX-2 still remains unknown now. Thus, we will do further study on NAB to identify the true reason why NAB did not suppress the COX-2 while the Nrf2 pathway is activated.

Point 2: The authors used dexamethasone as a positive drug for NO and PGE2. Although dexamethasone inhibit COX2, it does not belong to NSAID. The anti-inflammatory activity of DEX goes beyond COX inhibitors.

Response 2: Thank you for your valuable comment. It is true that DEX is not a kind of NSAID. The reason why we use DEX as a positive drug to evaluate the production of NO and PGE2 is that the DEX is a kind of classic anti-inflammatory drug widely used in clinic and it has been proved that the anti-inflammatory effect of it is strong. Thus, we selected the DEX as the positive drug to evaluate the anti-inflammatory effect of NAB in the first stage of the study. It is only to help us to identify whether the anti-inflammatory effect of NAB is strong and if it has the potential to be developed.

In the manuscript, we have discussed the reason why we selected DEX as the positive drug and the description is listed in red below here.

DEX is a kind of classic anti-inflammatory drug that is widely used in clinic [25]. It is a kind of steroidal anti-inflammatory drug that has been widely used to treat rheumatoid arthritic knees [26], pneumonia [27] and bronchiolitis [28].

Point 3: In Introduction, the authors should give a brief detail on NAB and the Nrf2/HO-1 pathway. Most of all, the relevance of Nrf2 to inflammation needs to be described.

Response 3: Thank you for this comment. We have added more description of Nrf2 to inflammation in the 1. Introduction part in the manuscript in red and it is also listed below:

The translocation of Nrf2 protein into the cell nucleus induces the expression of heme oxygenase (HO)-1. Then, followed by the overexpression of HO-1, the production of inflammatory mediators is reduced and the inflammatory process is modulated [14].

Point 4: It would be better to compare NAB with DEX and SFN together. It is confusing to use these reference compounds in different assays

Response 4: Thank you for the valuable comment. As explained previously, DEX in the study is to evaluate the strength of anti-inflammatory effect of NAB. And in the second stage of the study, the mechanism study of NAB, we selected SFN as the positive control instead of DEX because DEX is not a Nrf2 activator. We have compared the effect of NAB with DEX and NAB to the HO-1 expression, and the result was shown in Figure 4 (please check it in the cover letter in Word version we submitted). Taken together, the reason why we chose different positive drugs as reference is because the aim of the two stage of the study is different: in the first stage we need to evaluate the strength of anti-inflammatory effect of NAB; in the second stage we need to identify the mechanism of NAB to its anti-inflammatory effect.

Point 5: The authors only checked p-p65, p-p38 and p-ERK. p-JNK and IκBα degradation need to be verified

Response 5: Thank you for this comment. It is true that when we investigate the NF-κB pathway and the MAPK pathway, it is necessary to evaluate the degradation of p-JNK and IκBα. However, our study focused on the activation of Nrf2/HO-1 pathway. Also, the previously results showed that NAB did not depress the expression of p-p65, p-p38 and p-ERK. Thus, we did not do the experiments to the NF-κB pathway and the MAPK pathway deeper. Thanks for your suggestion, we will do some further studies on NAB to these proteins in the future studies.

Point 6: At least cytokine release should be evaluated using peritoneal macrophages

Response 6: Thank you for your comment. It is true that evaluate the cytokines release in peritoneal macrophages can verify the results we got now. According to the studies that have been reported and our experience, the results from the RAW 264.7 macrophages and the peritoneal macrophages are consistent in most situations. Moreover, the RAW 264.7 macrophages is a kind of macrophage cell line that is widely recognized in anti-inflammatory studies. Therefore, we did not use peritoneal macrophages in our study this time. 

Point 7: Figure 2D shows that NAB does not inhibit PGE2. Remove the following COX2 gene and proteins in Figure 2

Response 7: Thank you for this comment. Results from our pervious study showed that NAB does not inhibit PGE2. However, we considered that the tests on COX-2 gene and proteins are necessary. It is because that the COX-2 level influences not only the expression of PGE2 but also the synthesis of other PGs, such as PGH2, PGI2, PGD2 and TXA2. Some other researches have proven that the inhibition of COX-2 may break the balance between PGI2 and TXA2 and then causes cardiovascular risks. Thus, we listed the results of effect of NAB to the expression of COX-2. Also, we considered that the selective inhibition to iNOS instead of COX-2 is one of the advantages of NAB for the potential development.

Point 8: The qualities of several Western blot bands are poor. Replace figure 5 with the better one.

Response 8: Thank you for this comment. We have changed new Western blot images of Figure 5 and please check it in the cover letter in Word version we submitted.

Point 9: Lines 75-78 need to be relocated in the first and second paragraphs.

Response 9: Thank you for this comment. We have relocated those descriptions after the first paragraphs in red in the manuscript and it is also listed below:

Inflammation is a kind of defense reaction of living organisms with vascular system to the harmful factor such as pathogens, damaged cells, or irritants [1]. The inflammation may happen in numbers of diseases like arthritis, arthrophlogosis, asthma and so on [2]. The nonsteroidal anti-inflammatory drugs (NSAIDs) are widely used for fighting against inflammation diseases in clinical conditions. Restraining the cyclooxygenases (COXs) to inhibit prostaglandins is the main mechanism of NSAIDs to produce its anti-inflammatory effect [3]. However, many critical side effects, such as the increasing risk of serious, even fatal stomach and intestinal adverse reactions [4], myocardial infarction [5], stroke [6], systemic and pulmonary hypertension [7] and heart failures [8], happen while the COXs are inhibited. Therefore, NSAIDs is not that perfect for treating every inflammatory disease because of its side effects in clinic. Thus, it is necessary to develop new safer drugs to treat inflammation diseases better.

Macrophages plays important roles in the innate immune response. It protects cells from injury induced by exogenous factors like bacterial and virus and endogenous factors like other damaged cells. Also, macrophages promote the repair processes of tissue injury [9]. Pro-inflammatory mediators, such as interleukin-1β (IL-1β), interleukin -6 (IL-6), tumor necrosis factor alpha (TNF-α), prostaglandin E2 (PGE2), and nitric oxide (NO) [10-14], are produced by the activated macrophages and then promote the development of inflammation [15]. Thus, in our study, the LPS-stimulated RAW264.7 cells, a classical inflammatory cell model [16], was chosen to study the anti-inflammatory mechanism of NAB.

In recent years, there has been growing interest on the anti-inflammatory effects of natural components present in commonly used traditional herb medicine. Nardostachys chinensis is one of the traditional Chinese medicine that was reported to have anti-inflammatory effect [17]. The extracts of the plant roots and rhizomes of N. chinensis have been used for the treatment of blood disorders, disorders of the circulatory system and herpes infection [18]. Recently, some compounds isolated from N. chinensis were reported to inhibit the protein expressions of inducible nitric oxide synthase (iNOS) and cyclo-oxygenase-2 (COX-2) in LPS activated RAW264.7 macrophages [18-20]. Nardochinoid B (NAB) is a compound isolated from N. chinensis. Our previous research has proved that NAB inhibits the production of NO in the LPS-induced RAW264.7 macrophages [20]. However, the identification on the mechanisms of the anti-inflammatory action of NAB have not been done clearly. In this study, the mechanisms of anti-inflammatory activity and the antioxidant effect of Nardochinoid B (NAB) were for the first time investigated in LPS-stimulated RAW264.7 cells.

Point 10: Check Figure6A. It contains TNFa in microM

Response 10: Thank you for this comment. We have checked Figure 6A and changed the Y-Axis description in the manuscript and the cover letter we submitted.

I hope the above responses and the revised manuscript can satisfy the reviewers’ questions and comments. Should you have any queries, please do not hesitate to contact me at (+853) 8897 2458. Thank you again for consideration of our work. I look forward hearing from you.

Yours sincerely,

Hua Zhou, PhD

Professor

The State Key Laboratory of Quality Research in Chinese Medicine,

Macau Institute for Applied Research in Medicine and Health,

Macau University of Science and Technology

Taipa, Macau

huazhou2009@gmail.com

+853 88972458

Round 2

Reviewer 2 Report

Response 1: “All of the results in our study were analyzed by one-way ANOVA by using GraphPad Prism 7. We have added the description of data analysis in Materials and methods in red and listed below here. Moreover, all of the results in our study were evaluated and analyzed from three independent samples, which were gotten from three independent experiments. We believe that the “three independent experiments” in vitro is enough to prove our statements”.

Response to the authors:

I totally disagree that three independent experiments are enough to prove your statements. The one-way ANOVA can only be considered reliable if the samples are normally distributed. Your sample size is low to test for normal distribution. Another point, the one-way ANOVA is typically used to test for differences among at least three groups. I would recommend consulting a statistician.

Author Response

15th June 2019

Editorial Office

Molecules

Dear Editor,

Ref: Nardochinoid B inhibited the activation of RAW264.7 macrophages stimulated by lipopolysaccharide through activating Nrf2/HO-1 pathway (Revision 2)

Thank you and the reviewers for providing us the valuable comments and suggestions on our manuscript. According to these comments and suggestions, we have revised the manuscript and attach here the revision of our manuscript for your consideration of publication in Molecules. All the modified words or sentences have been incorporated in the revised manuscript and these modifications have been marked in red for ease of reference.

Reviewer #2:

Point 1: I totally disagree that three independent experiments are enough to prove your statements. The one-way ANOVA can only be considered reliable if the samples are normally distributed. Your sample size is low to test for normal distribution. Another point, the one-way ANOVA is typically used to test for differences among at least three groups. I would recommend consulting a statistician.

Response 1: Thank you for this comment.

First of all, we have added the description of data analysis in the previously revised manuscript in lines 404 to 408 as follow:

“All the data was presented as mean ± SEM for 3 individual experiments. Difference were analyzed by one-way ANOVA by using GraphPad Prism 7. In all cases, a level of P< 0.05 was considered statistically significant. One-way ANOVA was used in all statistical tests, the post-hoc analysis method was Tukey’s multiple comparison test.”

We think that our description may not be clear enough, so we have adjusted the description of data analysis like these in lines 405 to 407 in the newly revised manuscript:

“All data are presented as mean ± SEM of three independent experiments. The statistical analyses for these results were carried out with GraphPad Prism 7 by using One-Way ANOVA followed by post-hoc analysis with Tukey’s multiple comparison test to compare the difference between groups. In all cases, a level of P< 0.05 was considered statistically significant.”

Secondly, as we mentioned in our manuscript in line 60-62, the LPS-stimulated RAW 264.7 macrophages model is a classical inflammatory cell model that is widely approbated in the anti-inflammatory studies. And the study results from this cell model are stable and consistent. To prove our statement further, we have analyzed the distribution of the LPS-induced NO concentration in the RAW 264.7 cells, both the normal cells and stimulated cells produced in our lab recently, by using Shapiro-Wilk analysis with IBM SPSS Statistics 22.0. Twenty-five sets of samples were analyzed in the test and the results showed that the LPS-induced NO concentration in RAW 264.7 cells follows normal distribution, P (Sig.) >0.05. The results from IBM SPSS Statistics 22.0 is listed below.

Case Processing Summary

groups

Cases

Valid

Missing

Total

N

Percent

N

Percent

N

Percent

data

con

25

100.0%

0

0.0%

25

100.0%

lps

25

100.0%

0

0.0%

25

100.0%

Descriptive

groups

Statistic

Std. Error

data

con

Mean

1.8479

.09233

95% Confidence Interval for Mean

Lower Bound

1.6574

Upper Bound

2.0385

5% Trimmed Mean

1.8302

Median

1.7877

Variance

.213

Std. Deviation

.46165

Minimum

.87

Maximum

3.22

Range

2.35

Interquartile Range

.60

Skewness

.730

.464

Kurtosis

2.348

.902

lps

Mean

19.8980

.90735

95% Confidence Interval for Mean

Lower Bound

18.0253

Upper Bound

21.7707

5% Trimmed Mean

19.9757

Median

20.0735

Variance

20.582

Std. Deviation

4.53675

Minimum

12.07

Maximum

26.49

Range

14.42

Interquartile Range

8.41

Skewness

-.268

.464

Kurtosis

-1.173

.902

Tests of Normality

groups

Kolmogorov-Smirnova

Shapiro-Wilk

Statistic

df

Sig.

Statistic

df

Sig.

data

con

.112

25

.200*

.950

25

.254

lps

.168

25

.066

.923

25

.061

Furthermore, we have referenced and read many research papers that have been accepted and published by Journal of Experimental Medicine, Nature Immunology and Molecules, and we have found out that “results from three independent experiments” in the studies on the cell lines are acceptable.

In addition, we would like to point out that ANOVA was used to test for differences among at least three groups in the current research.

I hope the above responses and the revised manuscript can satisfy the reviewers. Should you have any queries, please do not hesitate to contact me at (+853) 8897 2458. Thank you again for consideration of our work. I look forward to hearing positive response from you soon.

Yours sincerely,

Hua Zhou, PhD

Professor

The State Key Laboratory of Quality Research in Chinese Medicine,

Macau Institute for Applied Research in Medicine and Health,

Macau University of Science and Technology

Taipa, Macau

huazhou2009@gmail.com

+853 88972458

Reviewer 3 Report

Some of the authors' response to my recommendation were not changed.

I do no agree with their responses.

I think that some results need to be changed.

Author Response

15th June 2019

Editorial Office

Molecules

Dear Editor,

Ref: Nardochinoid B inhibited the activation of RAW264.7 macrophages stimulated by lipopolysaccharide through activating Nrf2/HO-1 pathway (Revision 2)

Thank you and the reviewers for providing us the valuable comments and suggestions on our manuscript. According to these comments and suggestions, we have revised the manuscript and attach here the revision of our manuscript for your consideration of publication in Molecules. All the modified words or sentences have been incorporated in the revised manuscript and these modifications have been marked in red for ease of reference.

Reviewer #3:

Point 1: At least cytokine release should be evaluated using peritoneal macrophages.

Response 1: Thank you for the valuable comment. It is true that evaluating the cytokines release in peritoneal macrophages can verify the results we obtained. However, it is a pity that we do not have enough compounds to do any further research now. As we mentioned in the manuscript, compound NAB was isolated from Nardostachys chinensis. The isolation of NAB has been reported in our previous paper (Shen X Y, Qin D P, Zhou H, et al. Nardochinoids A–C, Three Dimeric Sesquiterpenoids with Specific Fused-Ring Skeletons from Nardostachys chinensis[J]. Organic letters, 2018, 20(18): 5813-5816.) Though we have isolated NAB from Nardostachys chinensis successfully, the extraction processes are complicated and the extraction rate of NAB was very low. So, in the previous research, we have already used all of the compound NAB to study its anti-inflammatory effect and no more compound left now. It is a regret that we cannot study further and do more experiments to the NAB in the short term because of the lack of the compound NAB. We agree that using peritoneal macrophages to test and verify our existing results is meaningful and beneficial. However, as we mentioned in our manuscript in line 60-62, the LPS-stimulated RAW 264.7 macrophages model is a classical inflammatory cell model that is widely approbated in the anti-inflammatory studies. And the study results from this cell model are stable and consistent. Thus, we consider that we are able to make positive and correct conclusions with our results from the RAW 264.7 cells. In the future, we will keep on isolating NAB from Nardostachys chinensis and do further research on it to enrich our study. Here we are politely to ask for your kind understanding that we are in very difficult situation to enrich our research in a short term.

I hope the above responses and the revised manuscript can satisfy the reviewer. Should you have any queries, please do not hesitate to contact me at (+853) 8897 2458. Thank you again for consideration of our work. I look forward to hearing positive response from you soon.

Yours sincerely,

Hua Zhou, PhD

Professor

The State Key Laboratory of Quality Research in Chinese Medicine,

Macau Institute for Applied Research in Medicine and Health,

Macau University of Science and Technology

Taipa, Macau

huazhou2009@gmail.com

+853 88972458